# Innovative Pedagogical Strategies in Health Professions Education: Active Learning in Dental Materials Science

**DOI:** 10.3390/ijerph20032041

**Published:** 2023-01-22

**Authors:** Galvin Sim Siang Lin, Wen-Wu Tan, Hung-Jia Tan, Chia-Wee Khoo, Kelvin I. Afrashtehfar

**Affiliations:** 1Department of Dental Materials, Faculty of Dentistry, Asian Institute of Medicine, Science and Technology (AIMST) University, Bedong 08100, Malaysia; 2Department of Dental Public Health, Faculty of Dentistry, Asian Institute of Medicine, Science and Technology (AIMST) University, Bedong 08100, Malaysia; 3Evidence-Based Practice Unit, Clinical Sciences Department, College of Dentistry, Ajman University, Ajman City P.O. Box 346, United Arab Emirates; 4Department of Reconstructive Dentistry and Gerodontology, School of Dental Medicine, University of Bern, 3010 Bern, Switzerland

**Keywords:** active learning, dental education, dental material, flipped classroom, group discussion, pedagogy, undergraduate

## Abstract

Dental materials science education is frequently delivered via traditional didactic lectures in preclinical dental programs. This review aimed to appraise the current evidence on innovative pedagogical strategies in teaching dental materials science courses. English-language articles on teaching methods for dental materials science published between January 1990 to October 2022 were searched in nine online databases (Google Scholar, PubMed, Web of Science [WoS], Science Direct, Cochrane Library, EBSCO, LILACS, Open Grey, and EMBASE) according to PRISMA guidelines. The risk of bias (RoB) was assessed using the Cochrane RoB-2 and ROBIN-I tools, whereas the level of evidence was determined based on the OCEBM guidelines. Only 12 primary studies were included. Two randomized studies (RCTs) were deemed as being of “some concern”, and one showed a high risk of bias (RoB). Three non-randomized controlled studies (NRS) demonstrated a moderate RoB, whereas the remaining seven were low. Most studies were ranked at Levels 2 and 3 of evidence. Several innovative pedagogical strategies were identified: flipped classrooms, clinical-based learning, computer-assisted learning, group discussion, microteaching with the BOPPPS (bridge-in, learning objective, pre-test, participatory learning, post-test, and summary) model, and game-based learning. The evidence suggested that students generally showed positive perceptions toward these pedagogical strategies. Dental educators should revise their current undergraduate dental materials science curricula and integrate more effective teaching methods.

## 1. Introduction

Most undergraduate dental curricula are primarily divided into preclinical and clinical phases, whereby students are taught the fundamental concepts of medicine and dentistry in the preclinical years prior to skill applications in the clinical years [1]. One of the essential preclinical courses is “dental materials science,” which applies concepts from materials science and chemical engineering to dentistry [2]. The course is sometimes perceived as “dry” due to its didactic pedagogical strategy with limited practical sessions [3]. Preclinical dental students may encounter obstacles in learning and comprehending dental materials science, as they mostly learn the course from a theoretical perspective [4]. Therefore, dental educators are interested in reconciling the vast quantity of dental materials knowledge they must impart to students to make them competent to perform later in the program’s clinical stage.

Typically, dental students are used to the traditional lecture format of teaching dental materials science courses, in which they engage passively by listening to the teacher, watching PowerPoint (PPT) slides and handouts, as well as taking notes during lectures [5,6]. Thus, students may have difficulty developing a profound and lasting comprehension in such huge lecture-based classes. In addition, the cost-effectiveness of giving a single lecture to a large class of students belies the fact that this educational approach frequently flops since it is challenging to promote higher-level cognitive activities and may run the danger of cognitive overload among students [7]. A change in dental materials science education has been advocated by shifting from conventional didactic learning to integrated student-centered learning with active engagement to promote self-directed learning [3].

The contemporary dental curriculum has experienced a paradigm shift from the traditional method of a discipline-based curriculum to an integrated competency-based curriculum [8]. Competency-based education outlines the knowledge, abilities, and ethical principles students must possess before practicing dentistry independently [9]. It is reasonable to argue that a student’s clinical competency is enhanced by the clinical application of dental materials [4]. Furthermore, students perceived that hands-on learning and peer discussion enhanced their understanding of dental materials science, which supports the need for innovative pedagogical strategies that encourage active learning [2]. The initiatives to combine competency-based dental education systems with new student-centered pedagogical strategies would provide students with a range of learning opportunities.

The higher levels of Bloom’s taxonomy of learning are stimulated when students actively participate in their learning, which encourages long-term memory and recall of the material [10]. Several studies have been conducted by incorporating new pedagogical strategies in dental material sciences courses to enhance the students’ understanding and learning experiences [3,11]. To the best of the authors’ knowledge, there is still a paucity of reviewed literature on pedagogical strategies for dental materials science. Hence, the present study aims to analyze and appraise the available evidence on innovative pedagogies in the 21st century for dental materials science to enhance current dental education.

## 2. Materials and Methods

The Preferred Reporting Items for Systematic Reviews and Meta-Analyses (PRISMA) guideline was followed for conducting the current evidence synthesis [12]. The review’s focused research question was:

“What are the current pedagogical strategies in dental materials science for undergraduate dental education?”

### 2.1. Search Strategy

On 3 November 2022, two review authors (CWK, HJT) independently conducted an electronic search using nine online databases (Google Scholar, PubMed, Web of Science [WoS], Science Direct, Cochrane Library, EBSCO, LILACS, Open Grey, and EMBASE) to search potential articles in the English language reporting dental materials pedagogical strategies published between January 1990 and October 2022. Furthermore, two additional researchers (GSSL, WWT) thoroughly examined reference lists of pertinent articles from the electronic search using a reference manager program (EndNote software, version X9, Thomson Reuters). A comprehensive database search was conducted using the Boolean operators “AND” and “OR” in conjunction with the first keywords (“teaching”, “pedagogy”, “pedagogical”, “instruction”, “learning”, “training”, “coaching”, “education”) and second keywords (“dental material”, “dental materials”, “dental biomaterial”, “dental biomaterials”, “dental materials science”, “dental materials sciences”). For example, “teaching” “AND” “dental materials” was used for the first search, and “teaching” “AND” “dental biomaterial” was used for the second search, and so on until all conjunctions of terms (e.g., MeSH, Emtree) and keywords were used. An example of the search using PubMed was listed in Table 1 and the other search strategies were adjusted accordingly based on each database.

### 2.2. Study Selection

Two researchers (GSSL, WWT) independently reviewed the records based on the title and the abstract after eliminating duplicate records using EndNote software version x9. Another two researchers (CWK, HJT) then carried out a full-text analysis to select relevant articles in accordance with the inclusion and exclusion criteria. The inclusion criteria were:(1)Randomized, non-randomized, cohort, case-control, or cross-sectional studies.(2)Presented in the English language.(3)Teaching methods for dental materials science courses.

In contrast, the exclusion criteria were:(1)Expert opinions, short communications, reviews, case reports, or case series.(2)Teaching methods for other dental-related courses.

Calibrations between researchers were conducted to determine inter-rater reliability. The average concordance was determined by utilizing the Kappa-value to compare the researchers’ decisions regarding inclusion and exclusion criteria [13]. All review authors discussed and resolved any disputes that arose during the research.

### 2.3. Data Extraction

Two researchers (CWK, HJT) used a modified excel spreadsheet extraction form to extract and document the parameters of each article. The following information was extracted: author, year of publication, country, type of study, sample size, pedagogical strategies, and general outcomes. Two review authors (GSSL, KIA) verified the accuracy of the data and convened a further discussion with all investigators if any discrepancies were found.

### 2.4. Risk of Bias Assessment

The risk of bias (RoB) for each included study was independently evaluated by three investigators (WWT, HJT, CWK) using two quality assessment tools. Randomized controlled studies (RCTs) were assessed using the Cochrane risk-of-bias tool for randomized trials (RoB 2) based on a five-domain of bias [14]. Either a “yes”, “probably yes”, “probably no”, “no”, or “no information” was assigned for each domain, and the studies were categorized as “low risk”, “some concerns,” or “high risk” based on the overall RoB judgment. The non-randomized controlled studies (NRS) were assessed using the Cochrane risk of bias in non-randomized studies of interventions (ROBINS-I) tool based on a seven-domain of bias [15]. Either a “yes”, “probably yes”, “probably no”, “no”, or “no information” was assigned for each domain, and the studies were categorized as “low risk”, “moderate risk”, “serious risk”, “critical risk”, or “no information” based on the overall RoB judgment. Additionally, the Oxford Centre for Evidence-Based Medicine (OCEBM) guideline was used to establish the level of evidence in each study [16]. Any discrepancies in study selection, data extraction, or quality assessment were resolved via discussion among all investigators until a consensus was achieved.

## 3. Results

### 3.1. Search Strategy

With a search timeframe spanning January 1990 to October 2022, the initial literature search turned up 12,956 articles (Figure 1). Following the removal of 7046 duplicate records, 5852 items were rejected based on their titles and abstracts, leaving 58 articles. The remaining papers underwent a comprehensive full-text evaluation based on the inclusion and exclusion criteria. Hence, only 12 primary studies were included in the present review. The average inter-rater Kappa score during the study selection procedure was 0.81 for the preliminary article screening (titles and abstracts) and 0.84 for the second screening (full-text evaluation), indicating a “strong” agreement [13].

Figure 1 also shows the reasons for the articles’ exclusion. Table 2 provides a summary of the characteristics of the included studies. Three included studies were RCTs [4,17,18], while the remaining nine studies were NRS [3,11,19,20,21,22,23,24,25]. Most primary studies were published in 2015 and beyond, except for one study published in 2008 [22]. Among the 12 studies, four originated from India, two from China, two from the United Kingdom, and the remaining four were from Pakistan, Germany, Turkey, and Romania. Figure 2 illustrates different pedagogical strategies available in the current evidence on dental materials science.

### 3.2. Risk of Bias Assessment

Table 3 shows the RoB for each included primary study. Based on the RoB 2 assessment tool, two RCTs were deemed as “some concern” [17,18], while one study showed a high RoB [4]. Meanwhile, three NRSs demonstrated a moderate RoB based on the ROBINS-I tool [20,21,23], whereas the remaining studies were classified as having a low RoB. Most included RCTs were ranked as Level 2, while all NRSs were ranked Level 3 based on the evidence of OCEBM.

### 3.3. Pedagogical Strategies

#### 3.3.1. Flipped Classroom

Flipped classroom (FC) is a new pedagogical strategy emphasizing learner-centered learning. In a flipped classroom approach, what is typically carried out in class and for homework is adjusted or “flipped”. For instance, students need to read some books or watch videos before attending class and engaging in active learning such as case studies and hands-on activities rather than sitting through a series of lectures in class or writing assignments at home. Two studies employed FC as an innovative pedagogical strategy in teaching dental materials science [18,20]. Joshi, J.P. et al. [18] revealed that students perceived the FC model to be better than the traditional classroom model. However, the knowledge assessments among students were comparable across both models. Hence, the study recommended that blended learning, which combines traditional classroom instruction with the FC model, be utilized to teach the course. Binnie, R.S.L. et al. [20] evaluated undergraduate dental students’ satisfaction with the applied dental materials and biomaterials course after the implementation of the FC. Most students expressed satisfaction with the FC delivery method, duration of teaching, material accessibility, and content understanding. Most students also concurred that the FC strategy was enjoyable and effective.

#### 3.3.2. Clinical-Based Learning

Two primary studies incorporated clinical-based learning into teaching dental materials science to undergraduate dental students [19,23]. Zhao, X. et al. [23] introduced the “integration-into-clinic” model as a new strategy in teaching dental materials science by dividing the curriculum into four teaching blocks in accordance with the clinical application of the dental material. According to the study, efficiency in both teaching and learning greatly improved. Furthermore, it was asserted that teachers had more time to devote to challenging and critical knowledge-based teaching, while students’ motivation in studying dental materials science and their aptitude for self-directed learning have both been enhanced through this innovative pedagogical strategy. Meanwhile, Qazi, H.S. et al. [19] compared the effectiveness of clinical rotation in teaching dental materials science courses with traditional lectures. It has been noted that there were higher levels of satisfaction and more favorable perceptions among students in the clinical rotation group. Compared to traditional lectures, students were more motivated and eager to learn about the course, and they could approach their dental materials practical work more confidently.

#### 3.3.3. Computer-Assisted Learning

Four studies evaluated the effect of computer-based technology and animation as pedagogical strategies in dental material sciences. Doloca, A. et al. [24] implemented computer-based Virtual Patient (VP) Software and determined the quality of virtual clinical cases, focusing on selecting dental materials. This strategy has successfully obtained a high level of acceptance among dental students. However, the degree of realism that VPs possess and the number of simulated scenarios should be improved. Meanwhile, another study introduced interactive digital experimental lectures on dental materials to substitute the conventional didactic lecture [21]. The technical implementation of the experimental lecture in the digital environment received positive feedback from the students. Additionally, the technical approaches to engage the class with padlets and lecture broadcasting were well received by the students.

Deshpande, S. et al. [25] demonstrated that students in the experimental group, following video and live demonstrations of relevant laboratory dental material procedures in addition to conventional lectures and guided laboratory tours, revealed a greater understanding of related topics than the control group according to their test scores. Furthermore, a previous study employed an electronic voting (eVoting) audience-response system in an undergraduate dental materials science course [22]. It was discovered that the eVoting system proved overwhelmingly accepted among students, but such an innovative pedagogical strategy was found to have no significant impact on the examination results.

#### 3.3.4. Group Discussion

Two studies integrated group discussion with didactic lectures in teaching dental materials science education. Gali, S. et al. [4] carried out case-oriented small group discussions (COSGDs) and showed that the perception of the students toward COSGDs is significantly better than the traditional classroom model. Most students felt group discussions help them to understand the subject better with discussions than lecture class alone. Students claimed they are more comfortable asking questions in a small group discussion set-up and felt they could relate the theory to clinical application better with a small group discussion with higher motivation to learn. Moreover, Sagsoz, O. et al. [17] employed a Jigsaw method in teaching dental materials science in which students were randomly divided into groups, studied their own topics, discussed interactively and then delivered the content to other members. The student performances were noted to improve with the Jigsaw method despite no significant difference from the traditional lecture method. However, the Jigsaw method was found to be more effective than lectures in students’ retention tests.

#### 3.3.5. Microteaching with BOPPPS Model

Yang, Y. et al. [3] used microteaching in conjunction with the BOPPPS method in the teaching of dental materials course. Students claimed that the teacher’s communication and explanations’ effectiveness were considerably higher. Moreover, students agreed that microteaching with the BOPPPS model encouraged them to participate more in the discussion, increased their interest in learning the course, enabled the teacher to encourage students to think independently, and helped them acquire and master the course material. In terms of overall academic achievement, it was found that students being taught using the microteaching and BOPPPS models outperformed those in the traditional method group.

#### 3.3.6. Game-Based Learning

Crossword puzzle-based stimulated games were introduced as a new pedagogy in dental materials science [11]. Most students agreed that crossword puzzles improved their ability to comprehend the material and enhanced their learning attitude. Students also perceived that the crossword puzzle helped them learn new words and remember important keywords. Furthermore, this new pedagogical strategy was deemed by most students to be an enjoyable and competitive learning activity.

## 4. Discussion

The present review appraised and addressed the current literature on innovative pedagogical strategies in dental materials science courses. The study of the teaching method is known as pedagogy. Based on the available evidence, traditional lectures are still a common approach in most dental curricula to impart knowledge in dental materials science. Lectures primarily involve a one-way style of communication that depends on passive learning rather than active student interaction [17]. Although lectures effectively disseminate fundamental knowledge and ideas, especially to large audiences, they neither assess students’ academic learning nor provide opportunities for students to take charge of their learning [10]. Finding evidence to justify a revision in dental materials science course design can be complicated since it involves different dental specialties content spanning preclinical and clinical phases [4,23]. Therefore, the design and development of innovative teaching strategies are required to change the way students learn in the 21st century.

The FC approach is an active learning strategy that encourages students’ engagement and places a focus on learner-centered instruction [10]. What is traditionally performed in class and as homework are swapped or “flipped” in an FC. This pedagogical strategy allows the subject matter to be more understandable and memorable through practical application in the classroom [10], makes teachers accessible to students as resources as they work through the subject matter [20], and promotes a collaborative environment with peer-to-peer learning and dynamic interactions [26]. For instance, rather than sitting through a lecture on impression materials in class and then completing assignments at home, students can have some reading materials before class, providing them with more opportunities to engage in their learning during classes. Through FCs, students could access the learning content and progress at their own pace. In addition, students can participate more fully in the succeeding lessons as a consequence of the excellent teacher-student contact time that followed from the FC approach [18,20]. Contrary to earlier research findings [27,28], the included study found no difference in knowledge assessments between the FC and traditional classroom approaches. The authors speculated that this could be attributed to the lack of motivation to complete the preparatory work among some students in the FC group [29].

Undeniably, applying classroom information to clinical settings is a significant hurdle in fundamental dental courses for students. By teaching the course in a clinical environment, several studies sought to close this gap and improve the learning experience for undergraduate students in dental materials science courses [19,23]. One way to enhance students’ learning is to revamp the course and integrate it into the dental curricula’s preclinical and clinical phases [30]. Teachers no longer need to spend much time explaining the clinical application and related clinical concepts if students have acquired the fundamental knowledge of the properties and manipulation of commonly used dental materials in the preclinical phase. This frees up more time for teachers to concentrate on teaching more crucial and challenging concepts. Nevertheless, the integration between the preclinical and clinical phases involves not only the challenge of combining theory and practice, but also the application of these competencies in practical settings with the complexities of actual patient care [31].

Digital technology integration into dental education has grown in popularity in recent years, particularly during the COVID-19 pandemic [32]. In fact, learners are becoming more confident participating in online classes after the pandemic since they can communicate with their peers flexibly and conveniently owing to computer-assisted online learning [33]. Mimicking actual patients and clinical circumstances using computer-assisted technologies can provide a more engaging, pleasant, and informative learning experience. According to the current review, dental students showed positive perceptions towards digital learning, such as virtual-patient software, e-voting audience-response system, and video demonstration, with encouraging feedback [22,24,34]. Unquestionably, as clinical exposure only occurs in the later years of most dental curricula, using digital technology to simulate clinical scenarios and select the appropriate dental materials is an effective strategy to improve students’ information assimilation and clinical reasoning [21]. Furthermore, digital technologies offer an option to teach dental materials science courses in schools with limited access to various dental materials. Nevertheless, dental students should be well-equipped with the skills to manage digital data and comprehend the advantages and drawbacks of digital procedures before such a pedagogical strategy can be fully implemented.

The current review highlights that group discussion improved students’ academic performance in dental materials science compared to traditional lectures [4,17]. Indeed, case-based small group discussions provide preclinical dental students with a chance to use their fundamental understanding of dental materials science to address problems connected to clinical dentistry practice [4]. In addition, Jigsaw learning was a more successful pedagogical strategy than lecture-based instruction as it enhanced students’ knowledge and ability to use clinical reasoning in dental materials science [17]. Small group discussion has also been proven to enable students to engage actively in their learning [5]. Students were more motivated to learn and felt more comfortable asking questions in small groups, as small-group peer learning created a sense of community among them [2,35].

Microteaching is a pedagogical strategy that involves teachers giving instruction in small groups and obtaining feedback from peers or mentors. Usually, a teacher is expected to impart a specific idea while utilizing a certain teaching technique [36]. The six phases of microteaching are plan, teach, feedback, replan, reteach, and re-feedback. This strategy involves sharing knowledge with a small group of peers for a brief period. After receiving criticism, the student reworks the lesson to address the issues, reteaches and obtains new feedback. It has also been reported that microteaching improved students’ learning activity [37]. Moreover, BOPPPS is a useful model for structuring classes. This includes piquing students’ interests, outlining the course objectives, evaluating their comprehension before class, engaging them in active learning, administering a post-test, and reviewing the material covered [3]. Compared to the conventional teaching strategy, the BOPPPS increased students’ enthusiasm and comprehensive ability [38].

Game-based learning is a significant advancement in healthcare education and a potent tool for active and problem-based learning [39]. Nonetheless, the practicality of knowledge transfer, player acceptance of the game, and learning through exploring new material are a few crucial components of this innovative strategy [40]. According to the present review, a crossword puzzle simulation game was utilized in teaching dental materials science. This allowed students to boost their knowledge confidence when they obtained the correct answer, increasing their sense of competence and satisfaction [11]. Laying out the effort to obtain the right answer might encourage students to participate actively in their learning. Previous research has proven that crossword puzzles have a positive effect on students’ ability to learn by analyzing theoretical concepts, memorization, learning terminology, and retrieving pertinent information while reducing the monotony of lectures and fostering a relaxed and friendly learning environment [41,42].

An assessment of the RoB in each primary study included in a systematic review is essential as it identifies any potential flaws in the compiled evidence and increases the credibility of the body of evidence [14]. The majority of the included NRSs were considered to have a low RoB in all domains except for the moderate RoB for the bias in the outcomes measured and selection bias in the reported result in three NRSs [20,21,23]. For instance, Zhao, X. et al. [23] did not specify how teaching and learning efficiency was measured or how many students had improved their ability for self-directed learning after implementing the “integration-into-clinic” model. In contrast, none of the three RCTs specified how the students were randomly allocated and did not clarify the proper randomization technique, although the term randomization was used in these articles [4,17,18]. However, it is understandable that blinding in selecting suitable students for such a social science study is difficult to achieve. In addition, no specific information was provided on missing data in Gali, S. et al. [4]. Moreover, there seemed to be discrepancies in the reported data.

The strength of the present review is that it offers significant insight into the field of dental education by contemplating various innovative teaching strategies to enhance students’ learning experience in dental materials science courses. It also paves the way for future researchers to perform a systematic review or meta-analysis of the chosen primary studies. The above-mentioned pedagogical strategies may somewhat overcome the limitations of traditional teaching methods, such as didactic lectures. The development of the cognitive capabilities required for the skilled practice of dentistry is related to active learning strategies that encourage critical thinking, engage in self-directed learning, and align with the different learning styles of the students [7]. Nevertheless, it is worth noting that one of the potential consequences of using students’ perceptions of their learning as a gauge for the effectiveness of these innovative pedagogical strategies is that positive attitudes may simply be the result of the strategies being a “novel” approach to learning, which is known as the Hawthorne effect [43]. Hence, students’ perceptions should not be regarded as the primary indicator of these pedagogical strategies’ efficacy [44,45]. In fact, they should be interpreted as a measure of the strategies’ acceptability and appeal to the students, which may impact their motivation to learn dental materials science.

Nevertheless, several drawbacks can be found in the current review. First, the present review did not identify sufficient primary studies with innovative teaching methods in dental materials science. However, the authors believed that including articles from a range of databases could provide dental educators with a better overall picture of the current research being conducted. Second, it is unavoidable that some pertinent articles might be missed, as studies published in languages other than English were excluded. In short, it is highly advocated that dental educators should conduct more well-controlled studies to address a variety of concerns, including teacher perceptions, teaching load, faculty training, availability of learning resources, students’ compliance with teaching activities, and other potential difficulties before these innovative pedagogical strategies can be successfully implemented in the existing curricula. Other ways to enhance students’ learning include curriculum mapping to avoid repeating dental materials science topics and integrating dental materials science courses into other dental specialities, such as teaching advanced CAD/CAM materials in prosthetic dentistry. In terms of assessment, it is also feasible to divide the students into two sections to measure the cognitive and psychomotor components.

## 5. Conclusions

The evidence suggested that students generally showed positive perceptions toward innovative pedagogical strategies in dental materials science that promote active learning as opposed to traditional lectures. These findings highlighted the necessity for future dental educators to revise their current undergraduate dental curricula and integrate effective teaching methods, notably in the dental materials science course. Prospective studies are also warranted to address concerns related to the implementation of these innovative pedagogical strategies and justify whether the adoption of these innovative strategies might result in an improvement in students’ academic performances.

## Figures and Tables

**Figure 1 ijerph-20-02041-f001:**
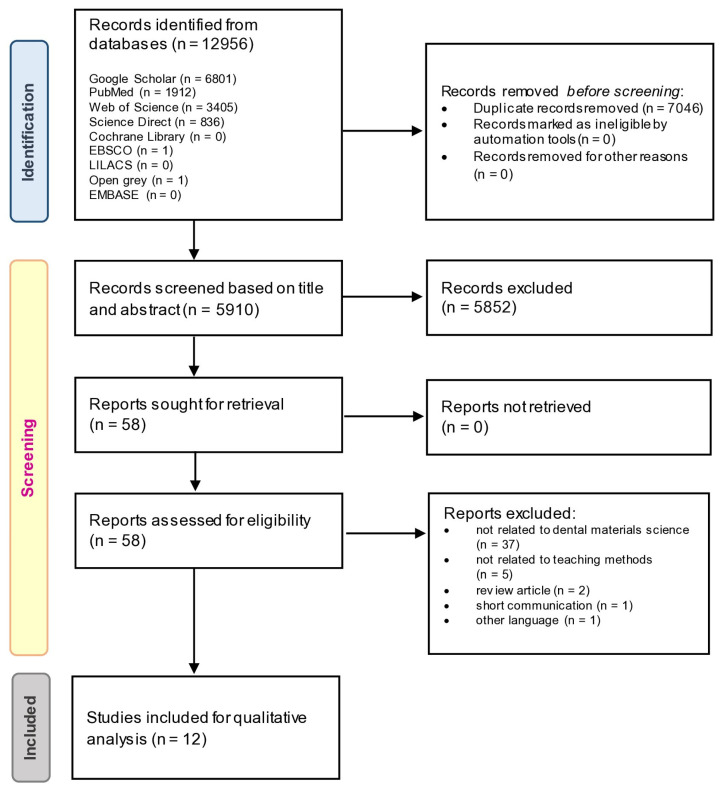
PRISMA flow diagram: number of records identified, included, and excluded, and the reasons for exclusions mapped out through the different phases of the current systematic review.

**Figure 2 ijerph-20-02041-f002:**
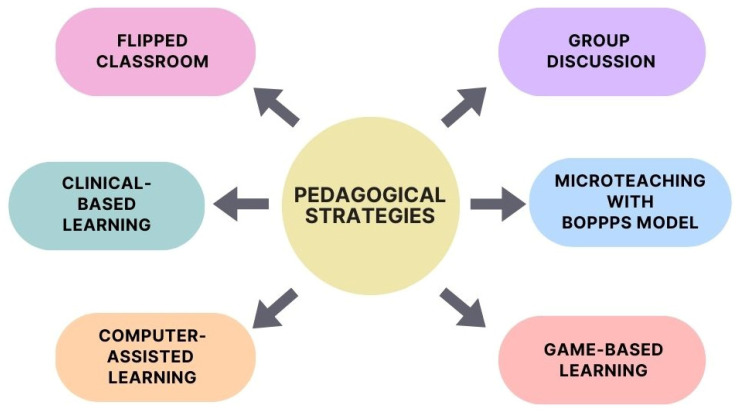
Different innovative pedagogical strategies in dental materials science education.

**Table 1 ijerph-20-02041-t001:** PubMed search strategy.

Search Strategy	
#1	((teaching AND dental material) OR (teaching AND dental materials) OR (teaching AND dental biomaterial) OR (teaching AND dental biomaterials) OR (teaching AND dental materials science) OR (teaching AND dental materials science))
#2	((pedagogy AND dental material) OR (pedagogy AND dental materials) OR (pedagogy AND dental biomaterial) OR (pedagogy AND dental biomaterials) OR (pedagogy AND dental materials science) OR (pedagogy AND dental materials science))
#3	((pedagogical AND dental material) OR (pedagogical AND dental materials) OR (pedagogical AND dental biomaterial) OR (pedagogical AND dental biomaterials) OR (pedagogical AND dental materials science) OR (pedagogical AND dental materials science))
#4	((instruction AND dental material) OR (instruction AND dental materials) OR (instruction AND dental biomaterial) OR (instruction AND dental biomaterials) OR (instruction AND dental materials science) OR (instruction AND dental materials science))
#5	((learning AND dental material) OR (learning AND dental materials) OR (learning AND dental biomaterial) OR (learning AND dental biomaterials) OR (learning AND dental materials science) OR (learning AND dental materials science))
#6	((training AND dental material) OR (training AND dental materials) OR (training AND dental biomaterial) OR (training AND dental biomaterials) OR (training AND dental materials science) OR (training AND dental materials science))
#7	((coaching AND dental material) OR (coaching AND dental materials) OR (coaching AND dental biomaterial) OR (coaching AND dental biomaterials) OR (coaching AND dental materials science) OR (coaching AND dental materials science))
#8	((education AND dental material) OR (education AND dental materials) OR (education AND dental biomaterial) OR (education AND dental biomaterials) OR (education AND dental materials science) OR (education AND dental materials science))

**Table 2 ijerph-20-02041-t002:** Characteristics of the selected primary studies.

No.	Author	Year	Country	No. of Participant	Study Design	Teaching Method	General Outcomes
1	Gali, S. et al. [4]	2015	India	170 UG students	RCT	Case-oriented small group discussions	Students felt that group discussions were more interesting, more comfortable asking questions, helped them understand the subject better, and provided a clearer picture for clinical correlation compared to traditional lecture classes.
2	Yang, Y. et al. [3]	2019	China	105 fourth-year PD students	NRCT (historically controlled study)	Microteaching combined with the BOPPPS model	Microteaching with the BOPPPS model can increase dental students’ enthusiasm and interest, promote critical thinking, and boost students’ academic performance as well as teaching quality.
3	Binnie, R.S.L. et al. [20]	2021	U.K.	N/A	NRCT (controlled before-and-after study)	FC	Most students were satisfied with the FC approach delivery strategy, instruction time allotted, information accessibility, and content understanding. Students also concurred that the FC was enjoyable and effective.
4	Qazi, H.S. et al. [19]	2019	Pakistan	101 first-year UG students	NRCT (historically controlled study)	Clinical rotation	Significant improvement in the clinical correlation of dental materials among students, along with higher satisfaction levels and more positive perceptions. Although it required more resources, the faculty found it a pleasant experience.
5	Saran, R. et al. [11]	2015	India	70 second-year UG students	NRCT (cross-sectional study)	Crossword puzzle	Most students claimed that crossword puzzles aided in effective teaching and comprehension of the topic, making the learning process more productive. They generally supported this teaching method in dental materials.
6	Kruppke, B. [21]	2021	Germany	N/A	NRCT (cross-sectional study)	Digital spaced-experimental lectures	The technical implementation of the experimental lecture in the digital space was rated very well among students. The technical methods for activating the students also received positive feedback.
7	Deshpande, S. et al. [25]	2018	India	60 second-year UG students	NRCT	Computer-assisted lecture + video and live demonstration	There was an improved understanding of related topics among students in the experimental group, evident from their test scores, and students perceived the teaching method positively.
8	Joshi, J.P. et al. [18]	2018	India	80 second-year UG students	RCT	FC	Despite that the assessment score of students in the FC model was superior to that of the traditional classroom model, there was no significant difference in knowledge content.
9	Barbour, M.E. [22]	2008	U.K.	142 second-year UG students	NRCT (historically controlled study)	Electronic voting	eVoting system had no significant impact on students’ examination results, but it received favorable perceptions among students, with greater learning engagement.
10	Sagsoz, O. et al. [17]	2015	Turkey	50 third-year UG students	RCT	Jigsaw learning	The pre-test and the post-test found no discernible difference between the lecture-based and Jigsaw methods, but the Jigsaw learning performed better in the retention test.
11	Zhao, X. et al. [23]	2018	China	N/A	NRCT (cross-sectional study)	“Integrated-Into-Clinic” model	Both teaching and learning efficiency increased significantly with teachers having more time to allocate the fundamental knowledge-based teaching, while students desired to learn and the competence for self-directed learning improved.
12	Doloca, A. et al. [24]	2016	Romania	73 third-year UG students	NRCT (cross-sectional study)	Virtual Patient (VP) software	The results indicated that VP could be an efficient tool for improving knowledge assimilation and clinical skills. This strategy received a good response and acceptance from the students.

UG: undergraduate; PG: predoctoral; NRCT: non-randomized controlled trial; RCT: randomized controlled trial; N/A: not available; FC: Flipped classroom.

**Table 3 ijerph-20-02041-t003:** Risk of bias and level of evidence of the selected primary studies based on the ROBINS-I, RoB 2 and OCEBM tools, respectively.

Component Study	Risk of Bias in Non-Randomized Studies of Interventions (ROBINS-I) Assessment Tool								Level of Evidence
	Bias Due to Confounding	Bias in Selection of Participants into the Study	Bias in Classification of Interventions	Bias Due to Deviations from Intended Interventions	Bias Due to Missing Data	Bias in Measurement of Outcomes	Bias in Selection of the Reported Result	Overall Risk	
Yang, Y. et al. [3]	Low	Low	Low	Low	Low	Low	Low	Low	3
Binnie, R.S.L. et al. [20]	Low	NI	NI	Low	Low	Moderate	Low	Moderate	3
Qazi, H.S. et al. [19]	Low	Low	Low	Low	Low	Low	Low	Low	3
Saran, R. et al. [11]	Low	Low	Low	Low	Low	Low	Low	Low	3
Kruppke, B. [21]	Low	NI	NI	NI	NI	Low	Moderate	Moderate	3
Deshpande, S. et al. [25]	Low	Low	Low	Low	Low	Low	Low	Low	3
Barbour, M.E. [22]	Low	Low	Low	Low	Low	Low	Low	Low	3
Zhao, X. et al. [23]	Low	Low	Low	Low	Low	Moderate	Moderate	Moderate	3
Doloca, A. et al. [24]	Low	Low	Low	Low	Low	Low	Low	Low	3
	**Revised Cochrane risk-of-bias tool for randomized trials (RoB 2)**								
		**Risk of bias arising from the randomization process**	**Risk of bias due to deviations from the intended interventions**	**Risk of bias due to missing outcome data**	**Risk of bias in measurement of the outcome**	**Risk of bias in selection of the reported result**	**Overall Risk**		
Gali, S. et al. [4]		Some concern	Low	High	Low	Low	High		2
Joshi, J.P. et al. [18]		Some concern	Low	Low	Low	Low	Some concern		2
Sagsoz, O. et al. [17]		Some concern	Low	Low	Low	Low	Some concern		2

NI: no information.

## Data Availability

No applicable.

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
