# Peer review of "Innovative Pedagogical Strategies in Health Professions Education: Active Learning in Dental Materials Science"

_ijerph, 2023, doi:10.3390/ijerph20032041_

Round 1

Reviewer 1 Report

The study is well conducted and article is nicely drafted. I congratulate the authors on their work.

Author Response

The authors sincerely thank the reviewer for the positive feedback after appraising our work. 

Reviewer 2 Report

The paper entitled “Innovative Pedagogical Strategies in Health Professions Education: Active Learning in Dental Materials Science“ is a systematic review analyzing the drawbacks of teaching dental materials during Dentistry education.

The paper is nicely written.

Nevertheless, I suggest some improvements.

Lines 88-96:

please add a table with one of the search strategies used (e.g., Pubmed) and reference to it in the text mentioning that the other search strategy was adjusted accordingly.

Line 98: At this stage, the articles are called records.

After the retrieval, they can be called papers or articles.

Lines 146-7:

The authors wrote:

Figure 1. PRISMA flow diagram: number of records identified, included, and excluded, and the 146 reasons for exclusions mapped out through the different phases of a systematic review. 

The reviewer thinks that the text should be modified: “through the different phases of the current systematic review.”

In 3.3.3 The authors could also cite another study that confirms the importance of online learning during the pandemic.

The authors could add a sentence like the following.

“Furthermore, it has also been reported that online e-learning has gained more and more popularity.” Reference: 

Paolone G, Mazzitelli C, Formiga S, Kaitsas F, Breschi L, Mazzoni A, Tete G, Polizzi E, Gherlone E, Cantatore G. 1 year impact of COVID-19 pandemic on Italian dental professionals: a cross-sectional survey. Minerva Dent Oral Sci. 2021 Dec 1. doi: 10.23736/S2724-6329.21.04632-5. Epub ahead of print. PMID: 34851068.

Author Response

Lines 88-96:

please add a table with one of the search strategies used (e.g., PubMed) and reference to it in the text mentioning that the other search strategy was adjusted accordingly.

A table has been added.

Table 1. PubMed search strategy.

Accordingly, a sentence referring to the new table has been added to the text.

Line 96-98:

An example of the search using PubMed was listed in Table 1 and the other search strategies were adjusted accordingly based on each different database.”

Line 98: At this stage, the articles are called records.

After the retrieval, they can be called papers or articles.

The authors agreed with the reviewer and changed the word “articles” to “records”.

Lines 146-7:

The authors wrote:

Figure 1. PRISMA flow diagram: number of records identified, included, and excluded, and the 146 reasons for exclusions mapped out through the different phases of a systematic review.

The reviewer thinks that the text should be modified: “through the different phases of the current systematic review.”

The phrase has been modified to comply with the reviewer’s suggestion.

In 3.3.3 The authors could also cite another study that confirms the importance of online learning during the pandemic.

The authors could add a sentence like the following.

“Furthermore, it has also been reported that online e-learning has gained more and more popularity.”

The authors agreed with the reviewer’s suggestion. However, the authors added the reference in the discussion section where computer-assisted learning was discussed.

Line 309:

“…particularly during the COVID-19 pandemic [32].”

Reviewer 3 Report

The article <Innovative Pedagogical Strategies in Health Professions Education: Active Learning in Dental Materials Science> requires some minor adjustments:

line 22

Three non-randomized controlled studies (NRS) demonstrated a moderate RoB. 

Please write “a moderate risk of bias”.

----

line 82

“On the first week of November 2022”

please indicate a defined date.

----

Section: Computer-Assisted Learning:

Please mention also the contribution of the Pandemic Covid-19 situation to the confidence of people in online lessons. People are more confident in learning online after the pandemic.

----

line 237-38 BOPPPS (bridge-in, learning objective, pre-test, participatory learning, post-test, and summary) 

The BOPPPS abbreviation was already defined in the abstrace, it should be removed

------

The authors should explain in detail what FC (flipped classrooms) are in the results. It is explained later at 267-73 in the discussion, but it should be in the results or in the Introduction.

------

The authors could also talk about several different strategies that came up in their mind to improve Dental Materials science teaching:

  1. lecture about the dental materials related to the subject being explained to students: E.g. cad/cam materials when prosthetic dentistry. Also talk about eventual disadvantages: repetitions (e.g.: impression materials, adhesives).
  2. split the teaching into 2 exams: the first more theoretical during the preclinic and a practical one afterward.
  3. etc.

Author Response

line 22

Three non-randomized controlled studies (NRS) demonstrated a moderate RoB.

Please write “a moderate risk of bias”.

The authors have changed the word “RoB” to “risk of bias”.

line 82

“On the first week of November 2022”

please indicate a defined date.

A defined date has been added.

Line 82:

On 3rd November 2022…

Section: Computer-Assisted Learning:

Please mention also the contribution of the Pandemic Covid-19 situation to the confidence of people in online lessons. People are more confident in learning online after the pandemic.

The suggested information has been added.

Line 309-312:

“In fact, learners are becoming more confident participating in online classes after the pandemic since they can communicate with their peers in a flexible and convenient manner owing to computer-assisted online learning [33]”

line 237-38 BOPPPS (bridge-in, learning objective, pre-test, participatory learning, post-test, and summary)

The BOPPPS abbreviation was already defined in the abstract, it should be removed

The words “bridge-in, learning objective, pre-test, participatory learning, post-test, and summary” have been removed.

The authors should explain in detail what FC (flipped classrooms) are in the results. It is explained later at 267-73 in the discussion, but it should be in the results or in the Introduction.

The authors have added the suggested information.

Line 183-188:

“Flipped classroom (FC) is a new pedagogical strategy that emphasizes learner-centered learning…rather than sitting through series of lectures in class or writing assignments at home.”

The authors could also talk about several different strategies that came up in their mind to improve Dental Materials science teaching:

lecture about the dental materials related to the subject being explained to students: E.g. cad/cam materials when prosthetic dentistry. Also talk about eventual disadvantages: repetitions (e.g.: impression materials, adhesives).

split the teaching into 2 exams: the first more theoretical during the preclinic and a practical one afterward.

etc.

Several points regarding the reviewer’s recommendation have been added.

Line 399-404:

“Other ways to enhance students' learning include curriculum mapping to avoid the repetition of dental materials science topics and integrating dental materials science course into other dental specialities, such as teaching advanced CAD/CAM materials in prosthetic dentistry. In terms of assessment, it is also feasible to divide into two sections to measure both the cognitive and psychomotor components.”
